# Wrist Movement Variability Assessment in Individuals with Parkinson’s Disease

**DOI:** 10.3390/healthcare10091656

**Published:** 2022-08-30

**Authors:** Lígia Reis Nóbrega, Ariana Moura Cabral, Fábio Henrique Monteiro Oliveira, Adriano de Oliveira Andrade, Sridhar Krishnan, Adriano Alves Pereira

**Affiliations:** 1Faculty of Electrical Engineering, Federal University of Uberlândia, Uberlândia 38400-902, MG, Brazil; 2Federal Institute of Brasília, Brasília 70070-906, DF, Brazil; 3Electrical and Computer Engineering, Toronto Metropolitan University, Toronto, ON M5B 2K3, Canada

**Keywords:** Parkinson’s disease, wrist movement, capacitive sensors, variability, quantitative evaluation, principal component analysis

## Abstract

(1) Background: Parkinson’s disease (PD) is a neurodegenerative disorder represented by the progressive loss of dopamine-producing neurons, it decreases the individual’s motor functions and affects the execution of movements. There is a real need to include quantitative techniques and reliable methods to assess the evolution of PD. (2) Methods: This cross-sectional study assessed the variability of wrist RUD (radial and ulnar deviation) and FE (flexion and extension) movements measured by two pairs of capacitive sensors (PS25454 EPIC). The hypothesis was that PD patients have less variability in wrist movement execution than healthy individuals. The data was collected from 29 participants (age: 62.13 ± 9.7) with PD and 29 healthy individuals (60.70 ± 8). Subjects performed the experimental tasks at normal and fast speeds. Six features that captured the amplitude of the hand movements around two axes were estimated from the collected signals. (3) Results: The movement variability was greater for healthy individuals than for PD patients (*p* < 0.05). (4) Conclusion: The low variability seen in the PD group may indicate they execute wrist RUD and FE in a more restricted way. The variability analysis proposed here could be used as an indicator of patient progress in therapeutic programs and required changes in medication dosage.

## 1. Introduction

Parkinson’s disease (PD) is a condition of the central nervous system (CNS), which affects the basal ganglia, causing progressive loss of dopamine-producing neurons in the substantia nigra [1,2]. This age-associated neurodegenerative disorder can lead to significant motor and non-motor disability [3].

PD is the second most prevalent neurological disorder. It affects individuals of different ages and epidemiological studies highlight an incidence of 17 cases per 100,000 people per year, with a higher incidence in men [4]. The reports of PD have shown an increased concern as it affects the patient’s quality of life [5,6].

A Parkinson’s Disease diagnosis is purely clinical and relies on medical history and neurological evaluation [7]. Clinically, the disease is characterized by four cardinal signs, i.e., bradykinesia, tremor, rigidity, and postural instability. Motor and non-motor dysfunctions are usually common during all stages of PD, although impairments become increasingly prevalent and obvious over the course of the disease [8]. The manifestations of Parkinson’s disease are diverse and occur in a heterogeneous way, with motor dysfunctions as the most well-known complications. Thus, reliable methods are needed to diagnose and assess the evolution of PD [9].

The need for a common and consistent method for the evaluation of PD led to the creation of the Unified Parkinson’s Disease Rating Scale (UPDRS) in 1984 [10]. The International Parkinson and Movement Disorder Society (MDS) revised and updated the scale in 2008, and it is now known as MDS-UPDRS [10,11]. The MDS-UPDRS has been the most widely used clinical scale for PD diagnosis and follow-up [12]. Since the last update of MDS-UPDRS, several technological advancements have been developed to aid in obtaining more precise information from patients with PD, including the addition of quantitative measures using various types of signal processing methods and sensors. Electromyography (EMG), electroencephalography (EEG), and inertial sensors are examples of typical quantitative measures [13,14,15], but these approaches require careful preparation of the skin, can cause skin irritation, and may not be used in situations with a high risk of contact contamination [7]. The capacitive measurement is an alternative solution for data collection without requiring contact with the skin and without restricting movement [16].

Correct monitoring is essential for Parkinson’s disease patients receiving appropriate treatment and follow-up. Qualitative approaches are typically used to assess motor dysfunctions in PD, for example, Part III of the MDS-UPDRS is retained as “motor examination”, which requires the training and experience of a health professional in a clinical setting. In this sense, a quantitative method would help to improve the evaluation of the individual, leading to more detailed information about the execution of a specific movement. Quantitative assessments provide more complete data but should be used in parallel with the clinical assessment. Quantitative evaluations are more useful for evaluating treatment strategies or describing movement than for diagnosing and preventing disease [9].

Among the quantitative measures, we can highlight the movement’s variability as the ability of the motor system to performing under a wide range of tasks and environmental constraints [17]. Literature defines two types of variability, coordinative and ‘end-point’, and they have opposing interpretations. Coordinative variability is defined as the variability of the interaction between segments or joints, whereas ‘end-point’ variability is defined as the variability of the product of a movement [18]. In 2002, a hypothesis put forward by Lipsitz [19] suggested that a lack of variability may be a characteristic of dysfunction in a performance, frailty, or disease. On the other hand, Hausdorff et al. [20] concluded that under usual walking conditions, step time variability is larger in a patient with PD, compared to healthy subjects. Hamill et al. [18] disclosed that the two types of variability are different, have different interpretations, and are related when goal-directed movements are examined.

Studies highlight the beneficial and adaptive aspects of variability in system function. From this perspective, increased variability is no longer rigidly associated with decreased skill levels, injury, and health [18,19]. In a clinical context, variability can be a source of information for the discrimination of patterns and the characterization of differences among studied populations [21]. Variability can be quantified using linear measures, such as magnitude variability, e.g., the average standard deviation along strides [22,23], or nonlinear measures, such as entropy that quantifies the structure of the temporal variability or regularity of a time series [23,24].

Several movements related to different joints and body segments can be used for objective measurements in people with PD, such as hip, knee, ankle, elbow, trunk, hand, finger, neck, and wrist movements [14,15,25].

Little is known about how wrist flexion and extension (FE) performance differs between people with Parkinson’s disease and healthy people, and how these movements affect other fine motor control tasks [26,27,28]. Wrist FE movements are acknowledged to play a significant role in the execution of several daily activities, such as taking a glass to the mouth, pouring from a pitcher, cutting with a knife, taking a fork to the mouth, using a telephone, reading a newspaper, and rising from a chair [26]. According to a review [29], wrist FE is an important activity for the assessment of motor signs in people with PD, even though it is not included in the MDS-UPDRS. From wrist FE, it is possible to estimate speed, amplitude, hesitations, interruptions, and decreases in the range of wrist motion, which are the main characteristics considered in the standard clinical evaluation.

Radial-ulnar deviations (RUD) are secondary movements for wrist FE, as well as for pronation and supination, which are movements present in Part III, motor examination, of the MDS-UPDRS [30]. It is known that RUD movements are necessary to achieve optimal flexion/extension and pronation/supination movements [31].

In most cases, PD initially affects the upper limbs of the individual [32]. Since the wrist is the joint responsible for coordinating hand movements, it has a direct impact on the performance of several important daily activities. Considering the subjectivity of hand movement assessment in Parkinson’s disease patients, the limitations of clinical scales, and the difficulties in performing a quantitative test in an appropriate laboratory, we propose a variability assessment of the hand movement execution in Parkinson’s disease patients using two pairs of non-contact capacitive sensors.

The need for this work stems from the fact that Parkinson’s disease still lacks a standard exam for diagnosis, follow-up, and treatment, and there is no agreement on whether it is better to have a single index for each parkinsonian symptom or if a set of variables is preferable [21]. Despite much research into the execution of hand movements in Parkinson’s disease, there are still problems to be solved and opportunities for innovation.

The proposed study is an objective evaluation of hand movement execution in Parkinson’s disease. The MDS-UPDRS has low internal consistency and agreement among evaluators for hand functions compared to full-scale scores [29]. Using an objective and quantitative evaluation, such as wrist movement assessment using capacitive sensors, could overcome these limitations.

The scale was last updated in 2008, and since then, various signal processing methods and sensors have been used to collect data from individuals with PD. The capacitive measure used in this study aims to provide an objective way to assess disease progression and evaluate care and therapy. Objective evaluation does not replace clinical evaluation but rather supplements it [9].

Considering that variability can be a source of information for pattern discrimination and characterization of differences among studied populations and that it refers to the ability of the motor system to performing in a wide variety of tasks, the goal of this study is to assess the variability of the wrist radial-ulnar deviation and flexion/extension execution measured in the electric field created by two pairs of capacitive sensors. The hypothesis is that people with Parkinson’s disease have less variability in their wrist movement execution than healthy people.

## 2. Materials and Methods

The study was conducted according to the guidelines of the Declaration of Helsinki, and all protocols were approved by the Ethics Committee of the Federal University of Uberlândia, Brazil. Informed consent was obtained from all subjects involved in the study.

Using two pairs of non-contact capacitive sensors, signals were collected from 29 healthy subjects (HS group, 60.70 ± 8.4 years) and 29 participants with Parkinson’s disease (PD group, 62.13 ± 9.7 years) [5,33]. Each group consisted of 20 men and nine women. Following the Shapiro-Wilk test for assessing the normality of the variables (*p* > 0.05), the Student’s *t*-test (*p* > 0.05) confirmed the statistical equality between the mean ages of the two groups (Table 1). Subjects were evaluated in laboratory research and the recruitment used a convenience sample.

### 2.1. Sensors

Non-Contact Capacitive (NCC) sensors were used to collect signals. NCC sensors are capable of measuring perturbations in the electric field induced by dielectric objects, such as the human body [33,34]. These sensors are based on electric potential sensing technology (sensor PS25454 EPIC, Plessey semiconductors, UK) and are capacitive sensors with ultra-high input resistance (~20 GΩ), input capacitance as low as 15 pF, lower −3 dB point typically of 0.2 Hz, typical upper −3 dB point of 20 kHz, and size of 10 mm × 10 mm. These features allow the detection of a disturbance in the electric field and the recognition of human activities due to the movement of a nearby object such as the hand.

For data collection, a 0.21 m × 0.21 m board with markers represented as targets (Figure 1) was created to assist the subject during wrist radial-ulnar deviations and flexion/extension; additionally, a laser was placed on the back of the subject’s hand with a micropore in order to facilitate the performance of experimental tasks. As shown by the arrows in Figure 1, the four sensors were placed at the four middle edges of a square area to create a field in which the hand can be inserted. Thus, each pair of sensors defined an axis, with the two perpendicular axes y (radial-ulnar axis) and z (proximal-distal axis). The acquisition system with a 2-D array of four NCC sensors PS25454 was validated by Oliveira et al. [7].

### 2.2. Tasks

Subjects performed ten tasks, described as follows:

Task 1 (T1)—Relaxed position

Task 2 (T2)—Pose against gravity with the laser pointed at the central region of the board (Figure 1)

Task 3 (T3)—Radial deviation

Task 4 (T4)—Ulnar deviation

Task 5 (T5)—Wrist flexion

Task 6 (T6)—Wrist extension

Task 7 (T7)—Radial deviation high speed

Task 8 (T8)—Ulnar deviation high speed

Task 9 (T9)—Wrist flexion high speed

Task 10 (T10)—Wrist extension high speed

Subjects completed the tasks while seated. They were instructed to place their hand in the system and begin the motor task in a relaxed position (T1) for 10 s, then switch to a pose against gravity (T2) with the laser pointed at the center of the board for 10 s. Figure 2 shows the flowchart of the study protocol.

Task 1 and task 2 were used solely to familiarize the subject with the signal acquisition system and were not included in the analyses. In addition, participants performed four wrist movement tasks, as shown in Figure 3: radial deviation, ulnar deviation, flexion, and extension. These tasks were used for analysis at normal and fast speeds.

Radial and ulnar deviation (T3, T4, T7 e T8) is the turning of the hand in the radial-ulnar direction (y-axis of the board) on the transverse plane and rotation around the proximal-distal axis with the following articular range: radial deviation from 0° to 15–25°, and ulnar deviation from 0° to 30–45°, as shown in the Figure 3b,c. Figure 3d,e demonstrate, respectively, flexion and extension (T5, T6, T9, and T10) of the wrist as the turning of the hand in the proximal-distal direction (z-axis of the board) on the coronal plane and rotation around the radial-ulnar axis, with the following articular range: flexion from 0° to 80–90° and extension from 0° to 70–90° [35]. Three trials were conducted to increase the number of observations, thereby contributing to a more reliable statistical result.

From task 3 to task 6, participants were required to perform the hand movements five times at normal speed per trial. A beep sounded every 2 s (auditory cue) for the subject to initiate movement, whereas from task 7 to task 10, they were required to perform the same movements as quickly as possible [34].

### 2.3. Signal Processing

Figure 4 shows the sequence for signal processing.

#### 2.3.1. Preprocessing

The signals detected by the capacitive sensor contain components of high frequency that are unrelated to voluntary movement. In order to obtain a waveform that best represents voluntary movement, it was necessary to eliminate these components. The linear trend of the signal was removed by fitting a linear model to the time series and then subtracting this trend from the data. The discrete wavelet transform was then used to decompose the signal into ten components. The employed wavelet was Daubechies of length 8. The lowest frequency component was selected as the component representing voluntary movement (Figure 5). The R packages *brainwaver* (Basic wavelet analysis of multivariate time series with visualization and parameterization using graph theory) and *pracma* (Practical Numerical Math Functions) were employed in the signal preprocessing stage.

#### 2.3.2. Windowing

The process of windowing was performed based on manual annotations. As shown in Figure 6, the initial and final times for each task were determined by observing the signal’s waveform. The *dygraphs* package in R was used to validate the correctness of the windowing process, where the function ‘dyShading’ was used to shadow the windowing index, and the ‘dyAnnotation’ function assisted in the visualization of the extracted features.

Figure 6 depicts a typical windowed signal for each hand movement task.

#### 2.3.3. Feature Extraction

The features were defined according to previous works [33,36,37,38], which collected attributes related to the amplitude of the signal. Based on this, we computed six features related to the amplitude. Table 2 shows the features used in this work, for I = 1, …, n, where n is the number of observations.

All the features were estimated for each task from the PD and HS groups. The variables were named based on the experimental conditions and feature extraction methods so that the task, movement execution speed, feature, and axis could be identified.

The tasks were identified as T = {T3, …, T10}, the features as F = {F1, …, F6} and the sensors as S = {S1, S2}. For instance, T4F1S1 represents the mean absolute value (F1) of the windowed capacitive signal during ulnar deviation (T4) along the y axis (S1), whereas T8F1S2 represents the estimate for the same feature (F1) during the maximum speed of ulnar deviation (T8) along the z-axis (S2).

#### 2.3.4. Standardization of Data

Data standardization is fundamental to facilitating and improving the use of data when comparing different features or physical parameters. Thus, the z-score was used for standardization of the variable as defined in Equation (1).
(1)$$\begin{array}{c} z=\frac{x-\mu }{\sigma }\; \end{array}$$
where x is the feature vector and μ and σ, respectively, correspond to the mean and standard deviation of x. Basically, the z-score measures how far a given value deviates from the mean in units of standard deviations. In general, this process of converting a raw score to a standard score is required for the use of multivariate data analysis and machine learning techniques because the models are often quite sensitive regarding the variances of variables.

#### 2.3.5. Bootstrap

For each variable, the overall mean coefficient of variability ($\bar{\mathrm{CV}}$) and the overall mean standard error ($\bar{\sigma_{\mathrm{CV}}}$) were estimated for each individual of the group by means of the use of the Bootstrap method for 1000 samples with replacement [39,40]. The coefficient of variability was calculated as follows in Equation (2):(2)$$\mathrm{CV}=\frac{\sigma }{\mu }\;$$
where σ is the standard deviation and µ is the mean of the samples of an individual in a group.

#### 2.3.6. Principal Component Analysis

To represent the pattern of similarity and reduce the dimensions of the data set, we applied the Principal Component Analysis (PCA), which reduces the data into its basic components [41], but minimizes information loss. This multivariate data analysis technique is based on data projection. Algebraically, data are represented through a set of linear transformations to preserve as much variability as possible and reduce redundancies. The goal is to represent the directions of the data that explain a maximal amount of variance in a K-dimensional space, using a set of orthogonal variables.

To decide how many principal components should be retained, it is common to summarize the results of a principal components analysis by using a scree plot, which we can do in R using the ‘screeplot’ function.

The results of PCA can be plotted on a biplot graph, which is a very popular way to view Principal Component Analysis results as it combines the main component scores and loading vectors in a single biplot screen. This allows us to quickly locate similar observations, clusters, outliers, and time-based patterns.

#### 2.3.7. Statistical Analysis

Initially, the normality of the coefficient of variation (CV) of each group was verified by the Shapiro–Wilk test with a significance level of 0.05 (*p* > 0.05). Student’s *t*-test was performed when data had a normal distribution, otherwise, the Mann-Whitney U test was performed to compare studied groups, both with a significance level of 0.05 (*p* < 0.05).

## 3. Results

In total, 96 TFS variables (8 tasks × 6 features × 2 axes) were estimated for each subject. Figure 7 shows the boxplot for the HS and PD groups of the T5F6S2, which represents the distribution of the values of the feature MAVSDN for motion task 5 (wrist flexion) in the z-axis.

Figure 8 shows the boxplot of the coefficient of variation estimated from each variable. The estimates were based on 1000 Bootstrap samples.

The scree plot was used to determine the number of principal components used. There is no objectively accepted method for determining how many principal components are required. This will be determined by the specific field of application and dataset. The Scree Plot, a plot of eigenvalues ordered from largest to smallest, is another method for determining the number of principal components. The number of components is determined when the remaining eigenvalues are all relatively small. As a result, the number of principal components used was determined in Figure 9.

According to Figure 9, 80.4% of the variability of the data is retained by the first two principal components for movements on the radial-ulnar axis (y). Similarly, 79.6% of the variability of the data is retained by the first two principal components for movements on the proximal-distal axis (z). Figure 10 depicts biplots of Principal Component Analysis for the most relevant variables in terms of explained variability for the principal components.

Correlated variables share the same direction. Positively correlated variables are located in the same quadrant, whereas negatively correlated variables are positioned in opposite quadrants of the graph origin. The cos2 values are employed to determine the quality of the representation. A high cos2 value indicates that the PCAs accurately represent the variable, and a low cos2 indicates that the PCAs do not perfectly represent the variable.

Table 3 lists, for the y and z axes, the seven and three (in gray) variables in Figure 10 that contribute the most. A table with the description of the variables that contribute the most to data variability is available on Appendix A.

Table 4 shows the results of coefficient of variability (CV) for the two groups considering the seven most relevant variables that contribute to data variability.

Figure 11 shows that the adopted method employing capacitive sensors can distinguish between PD patients and healthy older adults. The graph of individuals for radial-ulnar—y-axis (S1) and proximal-distal—z-axis (S2) depicts the dimension of the two first principal components for each axis, which represents the percentage of variances explained, and they are sufficient to distinguish between the two groups.

## 4. Discussion

Data analysis of the present study has delineated consistent results concerning wrist movement execution variability to properly differentiate the investigated groups. In all tasks and related features extracted, the PD group exhibited less variability compared to the HS group (Table 4).

The execution of the wrist movement could be captured by the two pairs of capacitive sensors, as shown in Figure 6, and the identification of the four hand movements (wrist radial-ulnar deviations and flexion/extension) does not require any additional nontrivial signal processing, for example, the peaks in the y axis, radial-ulnar direction, represent the capacitive sensor’s closeness during radial and ulnar deviation, and the peaks in the z-axis, proximal-distal direction, represent flexion and extension.

According to our results applying Principal Component Analysis (PCA), the combination of some features and tasks has higher relevance to different HS and PD groups, as demonstrated in Figure 10. After PCA, the variables that presented the greatest importance to discriminate the HS and PD groups were T6F4S1, T6F5S1, T10F1S1, T6F3S1, T10F4S1, T10F2S1, and T6F2S1 for y-axis; and T6F3S2, T6F5S2, T6F1S2, T6F2S2, T9F2S2, T5F2S2 and T10F2S2 for z-axis; as shown in Table 3.

According to PCA, task 5 and task 9 represent hand flexion and have high relevance to distinguishing HS from PD groups, but only in the proximal-distal axis. Furthermore, the findings suggest that, regardless of the evaluated axis, the extension movement (Task 6 and Task 10) is the most important for characterizing wrist movement and distinguishes HS from PD groups. It is known that the wrist joint has a key role in the kinetic chain that regulates hand movements and the wrist extensor muscles provide stability to this joint, increased grip strength, and optimal finger positioning [28,42], so the results are compatible with the literature.

Table 4 shows the coefficient of variability with greater values for healthy individuals than PD patients. Based on the results shown in Figure 8, it was possible to verify that all the investigated features were able to capture distinct variability measures, which considers the overall variability of the movement execution. These results are consistent with the hypothesis that a lack of variability may be a characteristic of dysfunction in a performance, frailty, or disease [19]. Variability influences the ability to control movement, therefore, the low variability seen in the PD group may indicate problems due to the lack of mapping of the sensory cortex with disturbances in the individual’s motor function [43]. Volunteers with Parkinson’s disease may think and concentrate to make the movement, once they have great difficulty performing learned movements automatically [44], therefore they do it in a more restricted way. Healthy people with full motor functions use several strategies to make the same movement.

The features that presented the greatest importance to discriminate the HS and PD groups were MAVFD, the mean of the absolute values of the first differences of x (F3), MAVSD, the mean of the absolute values of the second differences of x (F5), and MAV, the mean absolute value of x (F1), followed by RMS, root mean square (F2).

The manifestations of Parkinson’s disease are diverse and occur in a heterogeneous way [45], this highlights the need for an evaluation centered on the patient since correct monitoring is essential for PD patients to receive appropriate treatment and follow-up [9]. The subjectivity of the evaluation in PD remains nowadays, and Luiz et al. [46] proved that for a suitable evaluation in PD using the Movement Disorder Society—Unified Parkinson’s Disease Rating Scale (MDS-UPDRS), the golden-standard clinical scale, the health professional should have experience and internal consistency, what does not happens according to the literature [29].

Clinical evaluation of the hand movement has been included in the MDS-UPDRS [10], in items 3.4 finger tapping, 3.5 opening, and closing hand, and 3.6 pronation and supination of the hands, of Part III (MDS-UPDRS), the motor examination, hand movement assessment is performed by an evaluator who should rate the patient’s movement execution with a score from 0 (no problems) to 4 (unable or barely able to perform the task) [10]. The MDS-UPDRS has the advantage of being available to most clinicians, however, it requires the experience [46]. It is simple to identify extreme scores, 0 and 4, although, for scores 1, 2, and 3, the evaluator may have difficulty in accurately classifying the patient [47]. During the execution of the hand movement in which the patient performs the movement 10 times as fast and wide as possible, the evaluator scores each side separately, assessing speed, amplitude, hesitations, interruptions, and amplitude decrease. Additionally, following the MDS-UPDRS protocol, the patient should not repeat the movement series to solve doubts due to fatigue and familiarization.

When compared to full-scale scores, MDS-UPDRS Part III hand movements have lower values of internal consistency and agreement among evaluators [29]. Furthermore, it is important to note that Part III of the MDS-UPDRS contains 18 items in total, three of which are specific to the hand movement execution assessment. As a result, when it comes to assessing hand movement execution, the gold standard in Parkinson’s disease leads to a quick and superficial clinical evaluation.

The limitations of Parkinson’s disease follow-up have motivated the development of new methods for PD assessment [48]. These techniques are attempting to provide an objective way to measure the disease’s motor symptoms with the intention of improving the assessment of the disease progression and evaluating the effectiveness of care and therapy.

Therefore, a solution to improve the evaluation of PD is to combine clinical assessment with a quantitative measurement [28,49]. This study proposes a novel method for assessing wrist variability to differentiate between individuals with PD and healthy age-matched adults, using two pairs of capacitive sensors. One of the advantages of capacitive sensors is that they can be used to collect wrist movement data without direct contact with the skin.

Recent studies [28,50] show that a quantitative method would provide more complete data about the individual’s movements, e.g., the parameters collected by Rabelo et al. [28] could discriminate Parkinson’s disease patients from healthy older adults, proving that both inertial and EMG sensors are sensitive to the group’s differences while performing wrist extension against gravity with the forearm on pronation. Our findings show that capacitive sensors can also distinguish patients with PD from healthy older adults (Figure 11).

The discrimination between the groups, based on objective evaluation, may contribute to the accurate diagnosis of PD and to the monitoring of therapies [28]. The proposed method involves the analysis of four different movements of the wrist, RUD, and FE, by using capacitive sensors, on two axes, radial-ulnar (y-axis) and proximal-distal (z-axis) and the findings show that flexion and extension are the tasks that most distinguish the groups.

Six amplitude features, i.e., MAV, RMS, MAVFD, MAVFDN, MAVSD, MAVSDN, were chosen for analysis from the four movements and analyzed axes, to create a variable that represented the hand task, the amplitude of the movement, and the respective axis, and, from these features, to calculate variability using the coefficient of variation (CV) and compare the two groups (HS and PD). The variable representing the same task, feature, and axis could be seen for both groups, allowing the difference in variability between PD and HS to be visualized.

The movements radial ulnar deviation (RUD), represented in this study as task 3, task 4 (normal speed), task 7 and task 8 (high speed), and wrist flexion and extension (FE), represented in this study as task 5, task 6 (normal speed), task 9 and task 10 (high speed), have been used in numerous studies [28,30]. Recent studies have confirmed the relevance of wrist FE for the evaluation of PD motor signs [29], highlighting that alternative movements to those available in the MDS-UPDRS should be investigated. Among them, the use of distinct types of signal processing methods and sensors to collect data from patients with Parkinson’s disease can be studied further.

According to Corona et al. [49], the kinematic analysis of “hand-to-mouth” movement, particularly the reduced velocity and range of motion of elbow flexion-extension, is appropriate for representing upper limb movement alterations in people with Parkinson’s disease. The deviation from a physiological pattern allows for the tracking of disease progression or the effectiveness of pharmacologic and rehabilitative treatments. Our findings show the same reduced range of motion. In Figure 8, the boxplot of the coefficient of variation estimated from each variable and group shows that healthy individuals have a greater range of motion in variability than PD patients.

Muniz et al. [50] concluded that the variability can potentially be monitored as an indicator of patient progress in some therapeutic programs, and the proposed approach advances toward patient-centered monitoring.

Another important observation in Table 4 is that F4, the normalized mean of the absolute values of the first differences (MAVFDN) for extension is an outlier on the y-axis. T6F4S1 is the first principal component and T10F4S1 is the fifth principal component, according to PCA, and these are the only results in which the variability is greater for the PD group than for the HS group. According to Figure 10, T6F4S1 and T10F4S1 variables are negatively correlated with T6F5S1, T10F1S1, T6F3S1, T10F2S1 and T6F2S1 variables.

The outliers in T6F4S1 and T10F4S1 can be explained by considering F4 as a confounding feature. The wrist extension movement (T6 and T10) occurs in the proximal-distal direction, z-axis in the experiment. Conversely, when evaluating the same movement in the radial-ulnar direction, y-axis, the coefficient of variation is higher for the PD group. The interpretation of variability depends on the goal-directed movements and has adaptive aspects in system function [18]. Therefore, one can conclude that a healthy volunteer’s wrist is more stable in the radial-ulnar axis during wrist extension than PD volunteers.

Furthermore, the goal of this study was to contribute to the evaluation of Parkinson’s disease patients by providing a method for objectively measuring wrist movement and analyzing variability in movement execution using capacitive sensors. There is still room for technological innovation in Parkinson’s disease because we have not reached a point of agreement on whether it is better to have a single index for each parkinsonian symptom or a set of variables, and what is important to consider during the characterization of the motor signs.

Finally, because the control group defines the standard of normality and it is possible to measure how far an individual’s result is from normal, the variability analysis proposed here could potentially be used as an indicator of patient progress in therapeutic programs, such as physiotherapy treatment, or to monitor the change in medication dosage.

### Study Limitations

Any research on Parkinson’s disease progression depends on how we define the disease severity and the methods we use to measure severity and progression. The variability of the wrist tasks execution assessment with capacitive sensors can quantify the hand movement, specifically the extension movement. According to Regnault [51], when looking at the rate of PD progression, the key question is how we can best quantify the change (i.e., through imaging results, a physician-rated severity rating, and patient self-report). Therefore, the underlying issue of measuring the severity of PD is fundamental in this context.

## 5. Conclusions

This study described a non-contact sensing technology application that is completely passive and works to detect variations in the electrical field of the ambient. We assessed the variability of the wrist radial-ulnar deviations and flexion/extension execution measured in the electric field created by two pairs of capacitive sensors, using the coefficient of variation (CV). The wrist movement variability is greater for the HS group than for the PD group and it can help to discriminate between both groups. This confirms the hypothesis that people with Parkinson’s disease have less variability in their wrist movement execution than healthy people.

The features mean absolute value (first and second differences) extracted from wrist extension at normal and high speed are the combination most capable of illustrating the differences between the groups.

The methods used in this study can identify relevant landmarks for the progression monitoring of Parkinson’s disease, and they may be directly applicable to other neuromotor disorders. From a practical and clinical standpoint, the wrist movement assessment device based on two pairs of capacitive sensors could be used to extract a number of relevant parameters in hand movement execution.

## Figures and Tables

**Figure 1 healthcare-10-01656-f001:**
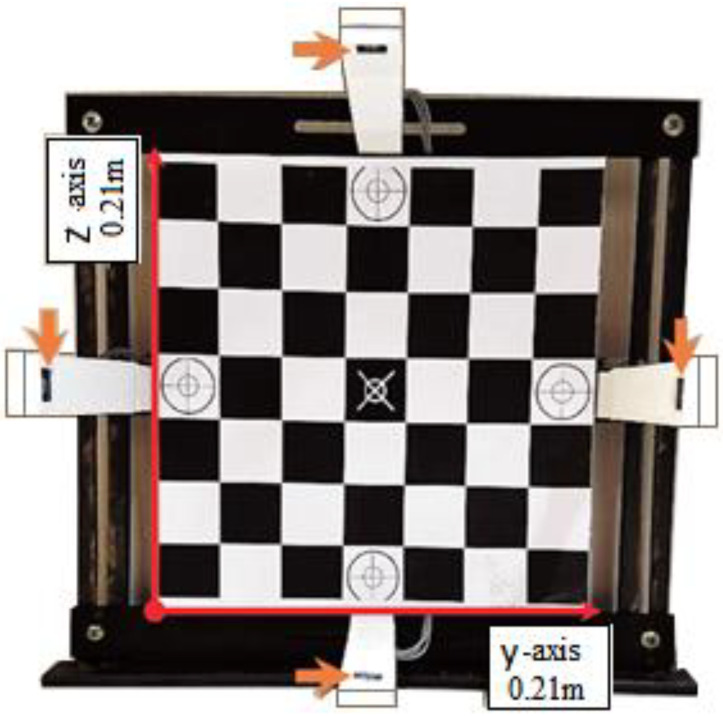
A 2-dimensional array of NCC sensors. Each sensor pair provides information regarding motion along the y and z axes, represented by the red arrows. A board with markers was placed in the background to aid the subject during the execution of experimental tasks. Orange arrows indicate the location of the four EPIC sensors.

**Figure 2 healthcare-10-01656-f002:**
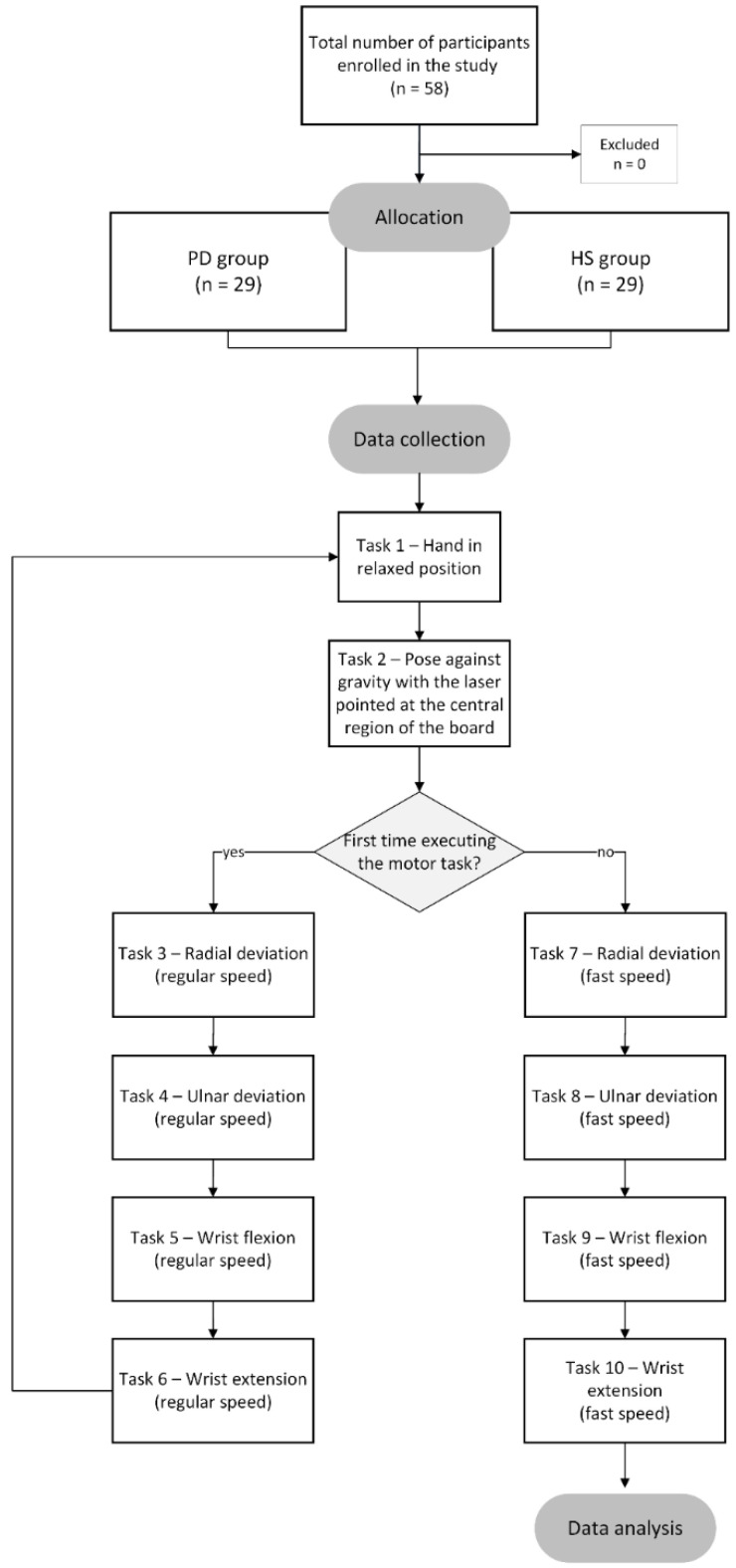
The study protocol shows volunteer allocation in PD and HS groups and presents the data collection protocol.

**Figure 3 healthcare-10-01656-f003:**
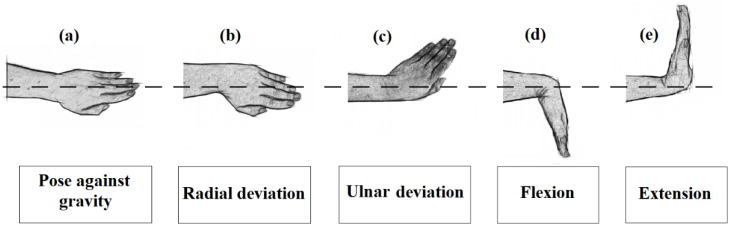
(**a**) hand in neutral position; (**b**) radial deviation; (**c**) ulnar deviation; (**d**) flexion and (**e**) extension.

**Figure 4 healthcare-10-01656-f004:**
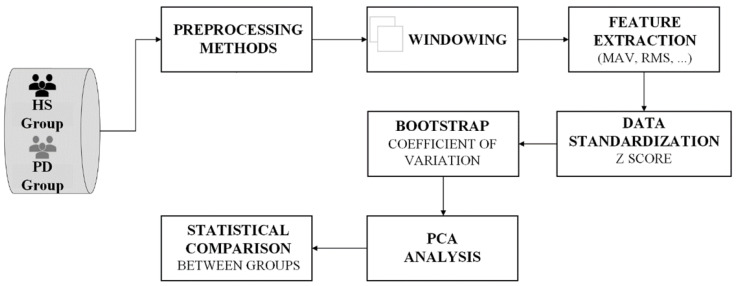
Flow diagram of signal processing.

**Figure 5 healthcare-10-01656-f005:**
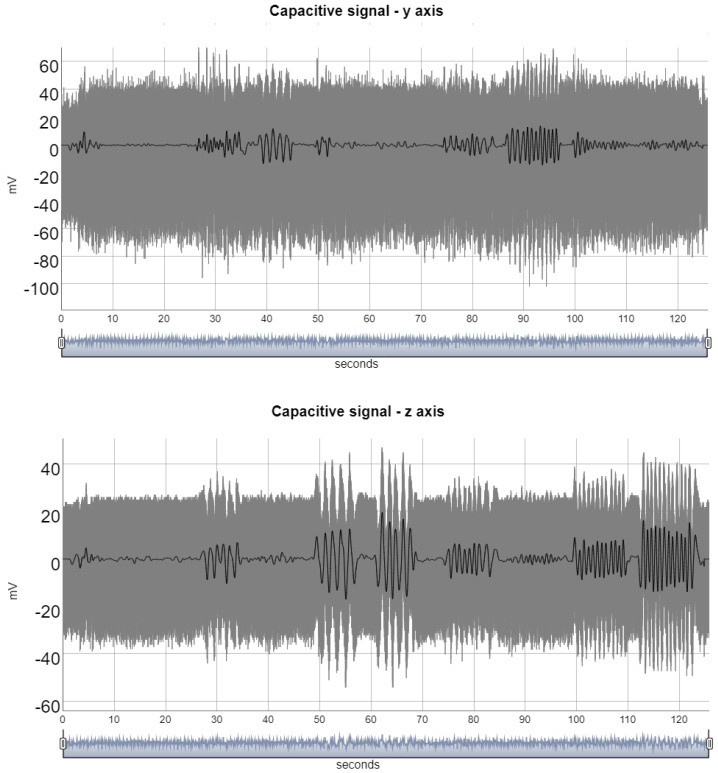
The graphics show the capacitive signal on the y and z−axis. The original signal is plotted in gray and the filtered signal, representing the voluntary movement, is plotted in black.

**Figure 6 healthcare-10-01656-f006:**
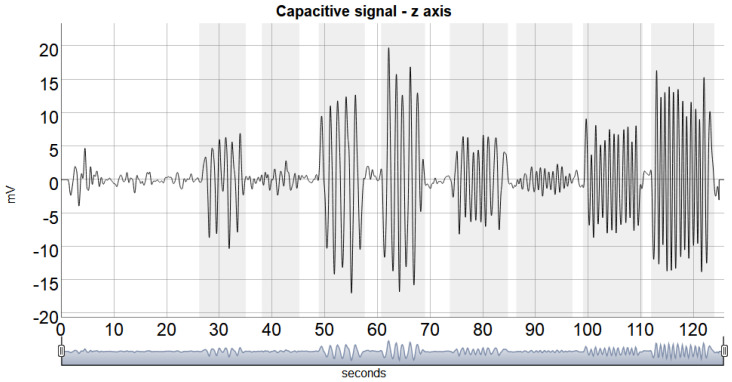
Typical hand movement detections A shadow indicates the tasks on the proximal-distal (z−axis).

**Figure 7 healthcare-10-01656-f007:**
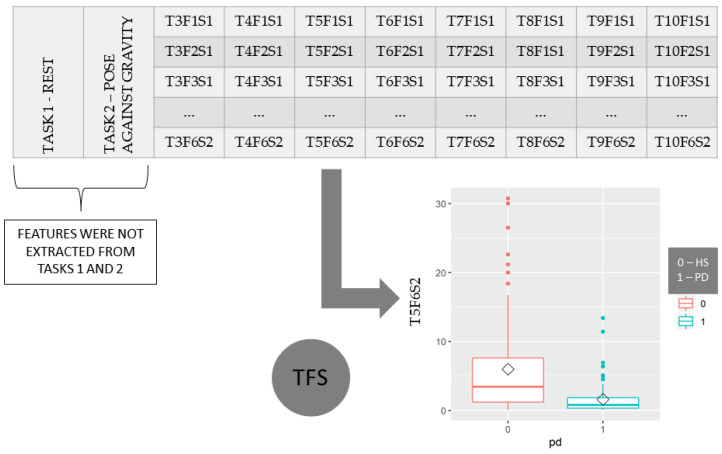
The table contains 96 TFS variables, and the boxplot on the right depicts T6F6S2 for the HS and PD groups.

**Figure 8 healthcare-10-01656-f008:**
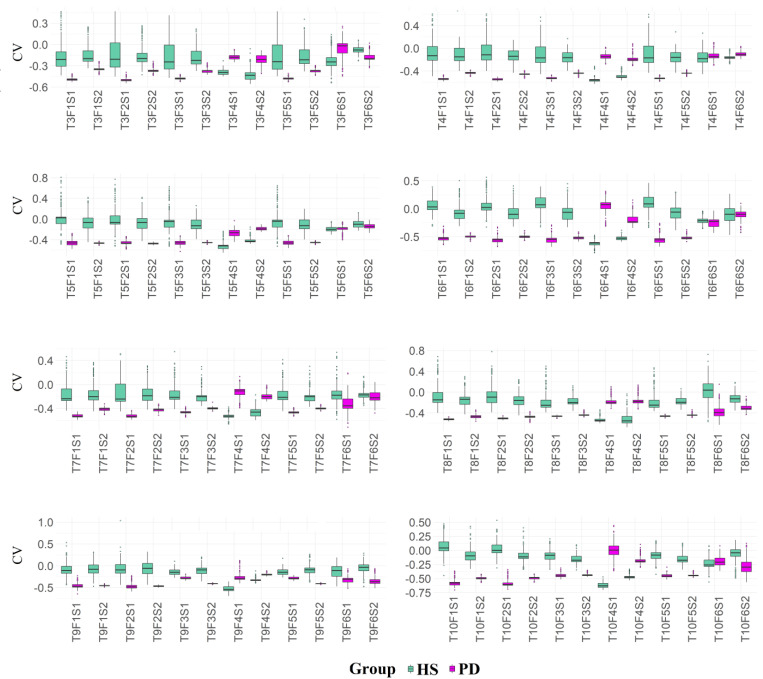
Coefficient of variation estimated for each variable and group (HS and PD). The estimates are based on 1000 Bootstrap samples.

**Figure 9 healthcare-10-01656-f009:**
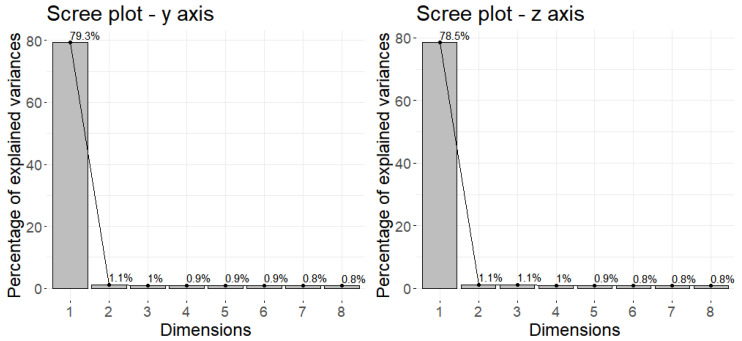
Scree plot for the radial-ulnar—y-axis (S1) and proximal-distal z-axis (S2).

**Figure 10 healthcare-10-01656-f010:**
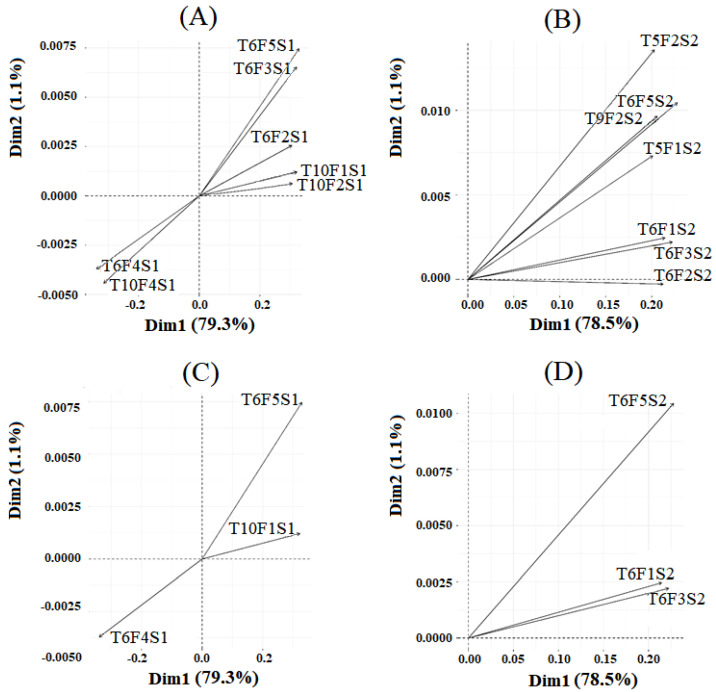
The top seven and three variables that contribute to data variability. (**A**) The top seven contributing variables to the radial-ulnar axis; (**B**) The top seven contributing variables to the proximal-distal axis; (**C**) The top three contributing variables to the radial-ulnar axis; and (**D**) The top three contributing variables to the proximal-distal axis.

**Figure 11 healthcare-10-01656-f011:**
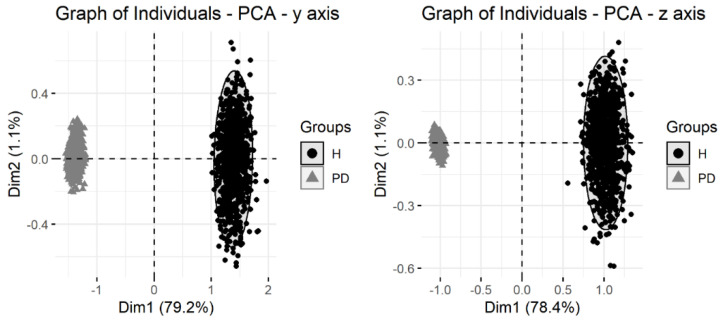
Graph of individuals for radial-ulnar—y−axis (S1) and proximal-distal—z−axis (S2).

**Table 1 healthcare-10-01656-t001:** Comparison between the mean age of PD and HS groups.

Group	Mean Age	Shapiro-Wilk	*p*-Value (*t*-Test)
PD	62.13	0.4614	0.5547
HS	60.70	0.2989	

**Table 2 healthcare-10-01656-t002:** Feature description and definition.

Feature		Description	Definition
F1	MAV	mean absolute value of the vector x	$\mathrm{MAV}=\frac{\sum_{i=1}^{n}|I|}{n}$
F2	RMS	root mean square of x	$\mathrm{RMS}=\sqrt{\frac{1}{n}\overset{n}{\underset{i=1}{\sum }}x_{i}^{2}}$
F3	MAVFD	mean of the absolute values of the first differences of x	$\mathrm{MAVFD}=\frac{1}{n-1}\overset{n-1}{\underset{i=1}{\sum }}{|x}_{i+1}-{\;x}_{i}|$
F4	MAVFDN	mean of the absolute values of the first differences of normalized x	$\mathrm{MAVFDN}=\frac{1}{n-1}\overset{n-1}{\underset{i=1}{\sum }}\frac{{|x}_{i+1}-{\;x}_{i}|-\overset{-}{x}}{\sigma }$
F5	MAVSD	mean of the absolute values of the second differences of x	$\mathrm{MAVSD}=\frac{1}{n-2}\overset{n-2}{\underset{i=1}{\sum }}{|x}_{i+2}-{\;x}_{i}|$
F6	MAVSDN	mean of the absolute values of the second differences of normalized x	$\mathrm{MAVSDN}=\frac{1}{n-2}\overset{n-2}{\underset{i=1}{\sum }}\frac{{|x}_{i+2}-x_{i}|-\overset{-}{x}}{\sigma }$

**Table 3 healthcare-10-01656-t003:** Seven and three variables (in gray) that contribute the most to data variability.

Axis	Variables						
**Y axis**	T6F2S1	T10F2S1	T10F4S1	T6F3S1	T10F1S1	T6F5S1	T6F4S1
**Z axis**	T10F2S2	T5F2S2	T9F2S2	T6F2S2	T6F1S2	T6F5S2	T6F3S2

**Table 4 healthcare-10-01656-t004:** Results of coefficient of variation for the two groups, with the *p*-value of the Shapiro-Wilk test and the *p*-value for the statistical test (*t*-test or Mann-Whitney U test).

TFS	T6F2S1			T10F2S1			T10F4S1			T6F3S1		
**Group**	CV	*p*-value	U	CV	*p*-value	U	CV	*p*-value	U	CV	*p*-value	U
**HS**	2.431	0.232	<0.001	4.316	0.548	<0.001	0.063	0.300	<0.001	1.225	0.314	<0.001
**PD**	0.074	0.024		0.083	0.009		40.100	<0.001		0.100	0.461	
**TFS**	**T10F1S1**			**T6F5S1**			**T6F4S1**			**T10F2S2**		
**Group**	CV	*p*-value	U	CV	*p*-value	U	CV	*p*-value	U	CV	*p*-value	*t*-test
**HS**	1.892	0.616	<0.001	1.1821	0.357	<0.001	0.073	0.919	0.001	0.978	<0.001	<0.001
**PD**	0.09	<0.001		0.100	0.161		1.817	0.002		0.048	<0.001	
**TFS**	**T5F2S2**			**T9F2S2**			**T6F2S2**			**T6F1S2**		
**Group**	CV	*p*-value	*t*-test	CV	*p*-value	*t*-test	CV	*p*-value	*t*-test	CV	*p*-value	U
**HS**	2.115	<0.001	<0.001	2.77	<0.001	<0.001	1.920	<0.001	<0.001	1.591	<0.001	<0.001
**PD**	0.026	<0.001		0.02	0.005		0.039	<0.001		0.035	0.037	
			**TFS**	**T6F5S2**			**T6F3S2**					
			**Group**	CV	*p*-value	*t*-test	CV	*p*-value	*t*-test			
			**HS**	1.779	<0.001	<0.001	1.762	0.002	<0.001			
			**PD**	0.039	0.002		0.038	<0.001				

## Data Availability

The datasets generated in the current study are not publicly available due to the ethical restrictions preventing the public sharing of data. A non-identified set may be requested after approval from the Review Board of the Institution. Requests for the data may be sent to the corresponding author.

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
