# Peer review of "Wrist Movement Variability Assessment in Individuals with Parkinson’s Disease"

_healthcare, 2022, doi:10.3390/healthcare10091656_

Round 1

Reviewer 1 Report

Parkinson's disease (PD) is a neurodegenerative disorder represented by the progressive loss of dopamine-producing neurons, it decreases the individual’s motor functions and affects the execution of movements. There is a real need to include quantitative techniques and reliable methods to assess the evolution of PD.

Authors  assessed the variability of wrist RUD (radial and ulnar deviation) and FE (flexion and extension) movements measured by two pairs of capacitive sensors (PS25454 EPIC). The hypothesis was that PD patients have less variability in wrist movement execution than healthy individuals. The data was collected from 29 participants (age: 62.13±9.7) with PD and 29 healthy individuals (60.70±8). Subjects performed the experimental tasks in a normal and fast speed. Six features which captured the amplitude of the hand movements around two axes were estimated from the collected signals.

Their study  showed that the movement variability was greater for healthy individuals than PD patients (p < 0.05).

The authors concluded that: (1) The low variability seen in the PD group may indicate they execute wrist RUD and FE in a more restricted way.(2) The variability analysis proposed here could be used as an indicator of patient progress in therapeutic programs and required changes in medication dosage.

This is an interesting article.

I have some minor suggestions with a pure academic spirit:

1. Add a clear purpose. In the intro I can see only the hypothesis

2. I can see a flow chart of the SP. I think that the introduction of a flow chart of all the design could improve the readability of the design.

3. Check the resolution of the figures (for example the figure 7).

4. There are  a  lot of acronyms. Pease minimize them and insert them in a table.

5. insert a recommendation in the conclusions

Reviewer 2 Report

The current manuscript is a cross- sectional study assess the Wrist Movement variability  in individuals with Parkinson’s Disease through sensors. It is informative, and we'll written. Please find my following comments: 

1- Lines 31-34 need to be revised.

2- Lines 81-84: out of context 

3- Report the study according to STROBE statement guidelines 

4- Lines 285-287 need to be revised 

5- Discussion: 

- start the discussion with short summary of the results

- There is no sufficient justification of the study results 

- There is no clinical or research implications

6- Conclusion: 

The conclusion is controversie the lines 70-74

Reviewer 3 Report

The study is well structured, elaborated and presented. Only the conclusion needs to be expanded. However, the resonance and importance of the subject dealt with is low in  my opinion

Round 2

Reviewer 2 Report

In the clinical implication, the authors proved the MDS-UPDRS for hand functions, so what is the importance of this study?
